# Food price trends during the COVID-19 pandemic in Brazil

**Giovanna Calixto Andrade**[1◉]*, **Thaís Cristina Marquezine Caldeira**[2◉], **Laís Amaral Mais**[3‡], **Ana Paula Bortoletto Martins**[1,4‡], **Rafael Moreira Claro**[2,5‡]

**1** Center for Epidemiological Research in Nutrition and Health (Nupens), University of São Paulo (USP), São Paulo, SP, Brazil, **2** Postgraduate Program in Public Health, Medical School, Federal University of Minas Gerais (UFMG), Belo Horizonte, MG, Brazil, **3** Institute for Consumers Defense (Idec), São Paulo, SP, Brazil, **4** School of Public Health, University of São Paulo (USP), São Paulo, SP, Brazil, **5** Nutrition Department, Federal University of Minas Gerais (UFMG), Belo Horizonte, MG, Brazil

◉ These authors contributed equally to this work.
‡ LAM, APBM and RMC also contributed equally to this work.
* gi.calixto.andrade@gmail.com

## Abstract

The present study aims to analyze the trends in food price in Brazil with emphasis on the period of the Covid-19 pandemic (from March 2020 to March 2022). Data from the Brazilian Household Budget Survey and the National System of Consumer Price Indexes were used as input to create a novel data set containing monthly prices (R$/Kg) for the foods and beverages most consumed in the country between January 2018 and March 2022. All food items were divided according to the Nova food classification system. We estimated the mean price of each food group for each year of study and the entire period. The monthly price of each group was plotted to analyze changes from January 2018 to March 2022. Fractional polynomial models were used to synthesize price changes up to 2025. Results of the present study showed that in Brazil unprocessed or minimally processed foods and processed culinary ingredients were more affordable than processed and ultra-processed foods. However, trend analyses suggested the reversal of the pricing pattern. The anticipated changes in the prices of minimally processed food relative to ultra-processed food, initially forecasted for Brazil, seem to reflect the impact of the Covid-19 pandemic on the global economy. These results are concerning as the increase in the price of healthy foods aggravates food and nutrition insecurity in Brazil. Additionally, this trend encourages the replacement of traditional meals for the consumption of unhealthy foods, increasing a health risk to the population.

## Introduction

An increase in the prevalence of overweight and obesity has been observed in developed and developing countries in recent decades, across all age groups [1]. In Brazil, in 2019, 60% of the adults were overweight, while 26% were obese [2]. Poor diet quality is one of the main causes of this scenario [3].

**Data Availability Statement:** This study used a secondary database that can be found in the Pesquisa de Orçamentos Familiares - POF/IBGE (https://www.ibge.gov.br/estatisticas/sociais/saude/24786-pesquisa-de-orcamentos-familiares-

2.html?=&t=microdados) and Sistema Nacional de Índices de Preços ao Consumidor- SIDRA/IBGE (https://sidra.ibge.gov.br/pesquisa/snipc).

**Funding:** Funding was received from: • Coordenação de Aperfeiçoamento de Pessoal de Nível Superior (Award number: 001) • Conselho Nacional de Desenvolvimento Científico e Tecnológico (Award number: 311170/2019-6) • International Development Research Centre (Award number: 109472-005) • Fundação de Amparo à Pesquisa do Estado de Minas Gerais (Award number: PPM-00325-17) The funders had no role in study design, data collection and analysis, decision to publish, or preparation of the manuscript.

**Competing interests:** The authors have declared that no competing interests exist.

Economic factors, such as price and income, stand out among the determinants of food consumption, especially among low-income individuals [4]. The price of a given food is directly associated with its consumption [5,6]. Therefore, many countries have adopted food price policies and actions aiming the promotion of healthy diets. The taxation of soft drinks adopted in countries such as Mexico and Hungary, for example, has been associated with reduced consumption of these beverages [7–9].

Considering the influence of price in food consumption, international organizations, such as World Health Organization (WHO) and Pan-American Health Organization (PAHO), recommend the taxation of unhealthy foods as an action to prevent non-communicable chronic diseases [7,8].

In Brazil, as national studies point out, the basis of the diet still consists of unprocessed or minimally processed foods that represent approximately 50% of the calories consumed [10]. The lower cost of unprocessed or minimally processed foods and processed culinary ingredients compared to ultra-processed food products (UPFP) has been one of the factors responsible for preserving the country's food culture [11].

Although the price of subgroups of unprocessed or minimally processed foods is not homogeneous (higher values per calorie for fruits, vegetables, meats; and lower values for cereals and pulses), the price per calorie of this group is still lower than the price per calorie of UPFP [11,12]. However, historical analyzes of food prices show a constant price increase in unprocessed or minimally processed foods and processed culinary ingredients compare to UPFP [12]. This trend has been observed since 1970s and coincides with the increase in the consumption of UPFP [10,13] and obesity [2] in Brazil.

A study based on food price data of the foods available in the Brazilian market from 1994 to 2017 forecasted price trends up to 2030 [12] and indicated that unhealthy diets based on UPFP could become cheaper than those based on unprocessed and minimally processed foods in 2026. However, the Covid-19 pandemic had a notorious impact on the global economy, and the food supply chain was directly affected [14]. As a result, food prices and price dispersion increased worldwide [15,16]. This scenario was also observed in Brazil [17,18]. Meanwhile, it is still unclear how this recent trend may have affected the previous forecasts [12].

The present study aimed to analyze the trends in food price in Brazil from January 2018 to March 2022 by assessing the period of the Covid-19 pandemic (from March 2020 to March 2022) and to projecting price scenarios until 2025.

## Methods

Data from the Household Budget Survey (*Pesquisa de Orçamentos Familiares*–POF 2017/2018) and from the National System of Consumer Price Indexes (*Sistema Nacional de Índices de Preços ao Consumidor*—SNIPC), both publicly available and collected by the Brazilian Institute of Geography and Statistics (*Instituto Brasileiro de Geografia e Estatística*—IBGE), were used to create a novel data set containing monthly prices (R$/kg) for the foods and beverages most consumed in the country (n = 95) between January 2018 and March 2022.

POF is carried out periodically (generally once a decade) and transversally, aiming to measure consumption structures, expenditures, and family income, identifying the profile of the population's living conditions [19]. POF was performed using a probabilistic sample, representative of the set of households in the country, and used a complex sampling plan by conglomerate, which involved geographic and socioeconomic stratification of all sectors in the country, with a drawing of census sectors in the first stage and households in the second. A detailed description of the sampling process is available elsewhere [20].

The SNIPC, implemented and managed by IBGE, continuously and systematically calculates the Consumer Price Index (*Índice de Preços ao Consumidor*—IPC). This index aims to identify the oscillation in the prices of goods and services related to the basket of goods in the Brazilian population. We defined the consumption baskets, with items consumed in the country, and the update of the IPC/SNIPC weighting structures through information from POF, which is carried out in the country, portraying the diversity of consumption habits observed throughout the Brazilian territory [19]. We used the Extended Consumer Price Index (*Índice Nacional de Preços ao Consumidor Amplo*—IPCA). The IPCA aims to measure the inflation of retail products and services related to the personal consumption of Brazilian with monthly incomes ranging from 1 to 40 minimum wages (an income range that guarantees the coverage of 90% of families belonging to urban areas covered by SNIPC, regardless of the source of income) [21].

The population standard basket of goods is organized into consumption categories of equivalent nature, including the "food and beverage" group. The collection of prices consists of a continuous task. The monthly calculation of the indexes is the result of the relative difference of the prices of each product in two consecutive months [19]. Then, we used the IPCA data (containing only the monthly price variation) from the SNIPC [19] to estimate annual prices from January 2018 to March 2022, based on 2017/2018 nominal values. Final price values have been adjusted to represent values through March 2022.

## Data organization

The SNIPC does not provide proper price data, only monthly price changes (by IPCA). Thus, we selected unit prices from POF 2017/2018 to calculate prices from 2018 to March 2022, using the monthly variation of the IPCA. We used the IPCA was used in the most disaggregated way possible. Data on 158 items, predominantly referring to a single product, were initially available. Of these, 63 were discarded due to insufficient data (price series available for a short period, less than five years or with missing data for several months).

Infusions (ground coffee and mate tea) and alcoholic beverages (beer, wine, and unspecified alcoholic beverage) were also excluded. Between infusions and alcoholic beverages, we observed a consumption pattern that differs from the other items on the consumption list. Specifically, alcoholic beverage purchases for home consumption are not recurrent. Finally, we included 95 items (foods or beverages) with complete information for the period (January 2018–March 2022).

Since the IPCA product list contains a smaller, more aggregated, and poorly described collection of items (which could encompass a variety of different foods, e.g., "cookies"), a qualitative process was carried out to determine the most appropriate POF 2017/2018 match for each of the 95 items on the IPCA list.

We performed a similar aggregation for the POF data for the aggregated items in the IPCA list. In cases where several POF items were considered suitable for an IPCA item, the most purchased product was used as the price series reference (e.g., "cracker"). This procedure was initially conducted independently by two researchers, and both lists were compared ($\kappa > 0.98$, excellent agreement). Disagreements were judged by a third researcher.

The Nova food classification system [22] was then employed. All food items were divided into four groups and their respective subgroups: 1) unprocessed or minimally processed foods (including meats, milk and eggs, vegetables, fruits, roots and tubers, cereals and pulses); 2) processed culinary ingredients (including vegetable and animal fats, sugar, and salt); 3) processed foods (including processed meats, processed vegetables, and processed bread); and 4) ultra-processed foods (including confectionery, sausages, cakes, bread, and crackers, soft drink, and other ultra-processed foods).

Considering that unprocessed or minimally processed foods are usually consumed with processed culinary ingredients, we calculated the price of these groups combined. Processed foods included canned or bottled fruits and vegetables; sweetened fruit pastes; salted or canned meats; canned fish; and artisanal breads. Flavored yoghurts; mayonnaise; biscuits; margarine; ice cream; chocolate and others were considered UPFP.

From the price of each product in January 2018 (from POF 2017/2018), the current price series (or "nominal price series") were calculated for each product using the formula:

$$A = B*(1 + (C/100)) \qquad (1)$$

Where A is the nominal food price in the current month, B is the nominal food price in the base month (or the nominal food price calculated for the previous month of the sequence), and C is the food price index in the current month [23].

The deflated price series (or 'actual price series') of each of these products was also calculated, using the following formula:

$$D = (E/F)*A \qquad (2)$$

Where D is the actual food price in the current month, E is the index number of the general food category in the base month (official price inflation data for specific categories, such as food, transport, health, education) [24], F is the index number of the general food category in the current month, and A is the nominal price in the month [23].

March 2022 was considered the base month for the calculation of the actual price series. The mean price of each food group and subgroup was estimated based on a weighted mean of the price of its constituents (the amount acquired (in kilograms) of each item according to POF 2017–2018). The weighting of each group was obtained by the updated weighting structures from the POF 2017/2018 [25].

## Statistical analysis

We estimated the mean price of each group and subgroup and the 95% confidence interval (95%CI) per year of study and the entire period. The monthly price of each group and its main subgroups were then plotted to analyze changes from January 2018 to March 2022. Fractional polynomial models were used to synthesize price changes up to 2025. Polynomials from the first to the fifth degree were evaluated. The polynomial two degree (largest R2) was chosen.

In addition, we calculated relative prices between healthy foods (unprocessed or minimally processed foods and processed culinary ingredients) and unhealthy foods (UPFP) based on the actual and estimated price series. The evolution of the accumulated growth rates of the IPC for the period between January 2018 and March 2022 was also observed based on the weights of food and monthly IPCA values obtained each month, whose weighting system was on a mobile basis (considering the immediately preceding period). The statistical software Stata (version 16.1) was used to organize and analyze the data.

## Results

Table 1 shows the price trend of food groups and subgroups from January 2018 to March 2022. In this period, UPFP were the most expensive group, followed by unprocessed or minimally processed foods and processed culinary ingredients, then processed foods. The food subgroups with the highest prices were processed meats, confectionery and minimally processed meats, and the lowest prices were for salt, sugar, and soft drink. From January 2018 to March 2022, a price increase in unprocessed or minimally processed foods and processed culinary ingredients was observed (from R$ 15.3/kg in 2018 to R$ 18.1/kg in March 2022). The price of

**Table 1. Mean prices (with 95%CI) of unprocessed or minimally processed foods and processed culinary ingredients, processed and UPFP from January 2018 to March 2022.** Brazil, 2018–2022.

| Food group/subgroup | Price (R$/kg) | | | | | | | | | | | |
|---|---|---|---|---|---|---|---|---|---|---|---|---|
| | 2018 | | 2019 | | 2020 | | 2021 | | 2022* | | Total | |
| | Mean | 95% CI | Mean | 95% CI | Mean | 95% CI | Mean | 95% CI | Mean | 95% CI | Mean | 95% CI |
| **Unprocessed or minimally processed foods and processed culinary ingredients** | 15.11 | 15.02 - 15.19 | 15.23 | 14.86 - 15.61 | 16.46 | 16.19 - 16.73 | 17.38 | 17.24 - 17.52 | 17.43 | 16.02 - 18.83 | 16.13 | 15.83 - 16.42 |
| *Unprocessed or minimally processed foods* | | | | | | | | | | | | |
| Meats | 24.46 | 24.27 - 24.66 | 24.67 | 23.57 - 25.77 | 27.38 | 26.64 - 28.13 | 29.95 | 29.60 - 30.30 | 29.85 | 27.21 - 32.49 | 26.81 | 26.09 - 27.53 |
| Milk and eggs | 12.06 | 11.81 - 12.30 | 11.49 | 11.39 - 11.58 | 11.54 | 11.41 - 11.68 | 11.33 | 11.23 - 11.43 | 11.22 | 10.02 - 12.42 | 11.58 | 11.47 - 11.69 |
| Vegetables | 12.38 | 11.97 - 12.78 | 13.16 | 12.59 - 13.74 | 13.81 | 13.10 - 14.52 | 12.81 | 12.65 - 12.97 | 14.09 | 12.15 - 16.04 | 13.10 | 12.83 - 13.37 |
| Fruits | 6.79 | 6.66 - 6.91 | 6.95 | 6.83 - 7.08 | 7.23 | 7.10 - 7.36 | 6.91 | 6.61 - 7.20 | 7.30 | 7.13 - 7.47 | 6.99 | 6.90 - 7.08 |
| Cereals and pulses | 5.38 | 5.35 - 5.41 | 5.46 | 5.34 - 5.58 | 5.76 | 5.54 - 5.97 | 5.90 | 5.71 - 6.09 | 5.32 | 5.25 - 5.38 | 5.61 | 5.52 - 5.70 |
| Roots and tubers | 4.13 | 3.76 - 4.49 | 5.75 | 5.15 - 6.36 | 5.31 | 4.76 - 5.86 | 4.92 | 4.44 - 5.40 | 4.89 | 4.80 - 4.99 | 5.02 | 4.75 - 5.29 |
| *Processed culinary ingredients* | | | | | | | | | | | | |
| Vegetable and animal fats | 13.84 | 13.69 - 13.99 | 13.49 | 13.40 - 13.59 | 13.90 | 13.21 - 14.59 | 15.29 | 15.14 - 15.44 | 15.62 | 14.26 - 16.98 | 14.22 | 13.95 - 14.48 |
| Sugar | 3.00 | 2.93 - 3.07 | 2.92 | 2.87 - 2.96 | 3.00 | 2.97 - 3.03 | 3.42 | 3.23 - 3.62 | 3.84 | 3.69 - 3.98 | 3.13 | 3.04 - 3.22 |
| Salt | 2.29 | 2.27 - 2.30 | 2.22 | 2.19 - 2.25 | 2.09 | 2.03 - 2.14 | 1.92 | 1.90 - 1.93 | 1.87 | 1.83 - 1.91 | 2.11 | 2.07 - 2.16 |
| Other ingredients | 16.55 | 16.45 - 16.64 | 15.97 | 15.80 - 16.14 | 15.02 | 14.76 - 15.28 | 14.06 | 13.87 - 14.25 | 14.14 | 11.69 - 16.58 | 15.33 | 15.03 - 15.62 |
| **Processed foods** | 16.82 | 16.72 - 16.91 | 16.36 | 16.24 - 16.47 | 15.96 | 15.70 - 16.22 | 15.16 | 15.00 - 15.32 | 14.96 | 13.88 - 16.04 | 16.01 | 15.81 - 16.21 |
| Processed meats | 34.16 | 33.93 - 34.38 | 33.39 | 32.85 - 33.93 | 35.84 | 35.61 - 36.07 | 35.42 | 35.01 - 35.84 | 34.68 | 30.49 - 38.87 | 34.70 | 34.38 - 35.03 |
| Processed vegetables | 21.93 | 21.75 - 22.11 | 20.98 | 20.77 - 21.18 | 19.89 | 19.62 - 20.16 | 18.84 | 18.71 - 18.96 | 19.00 | 16.90 - 21.10 | 20.33 | 19.98 - 20.67 |
| Processed bread | 12.72 | 12.58 - 12.86 | 12.42 | 12.32 - 12.52 | 11.68 | 11.40 - 11.95 | 10.86 | 10.73 - 10.99 | 10.68 | 10.33 - 11.02 | 11.85 | 11.62 - 12.07 |
| **Ultra-processed food** | 21.78 | 21.61 - 21.96 | 20.50 | 20.26 - 20.74 | 19.27 | 18.99 - 19.56 | 18.50 | 18.39 - 18.61 | 18.60 | 16.81 - 20.38 | 19.93 | 19.56 - 20.30 |
| Confectionery | 41.87 | 41.37 - 42.38 | 38.13 | 37.29 - 38.98 | 35.10 | 34.51 - 35.69 | 32.99 | 32.80 - 33.18 | 33.64 | 27.55 - 39.73 | 36.83 | 35.84 - 37.81 |
| Sausages | 20.51 | 20.30 - 20.72 | 19.69 | 19.55 - 19.82 | 19.09 | 18.95 - 19.24 | 19.45 | 19.30 - 19.59 | 19.07 | 17.75 - 20.39 | 19.65 | 19.48 - 19.81 |
| Cakes, bread and crackers | 20.14 | 19.99 - 20.30 | 19.07 | 18.91 - 19.23 | 17.97 | 17.59 - 18.35 | 16.74 | 16.58 - 16.90 | 16.83 | 15.79 - 17.87 | 18.38 | 18.00 - 18.76 |
| Soft drink | 4.68 | 4.64 - 4.73 | 4.55 | 4.51 - 4.59 | 4.25 | 4.15 - 4.35 | 3.88 | 3.85 - 3.91 | 3.80 | 3.74 - 3.87 | 4.31 | 4.21 - 4.41 |
| Other ultra-processed foods | 15.37 | 15.28 - 15.45 | 14.89 | 14.74 - 15.04 | 13.80 | 13.53 - 14.07 | 13.43 | 13.32 - 13.54 | 13.76 | 13.10 - 14.43 | 14.34 | 14.10 - 14.57 |

*Until March 2022.

CI: confidence interval.

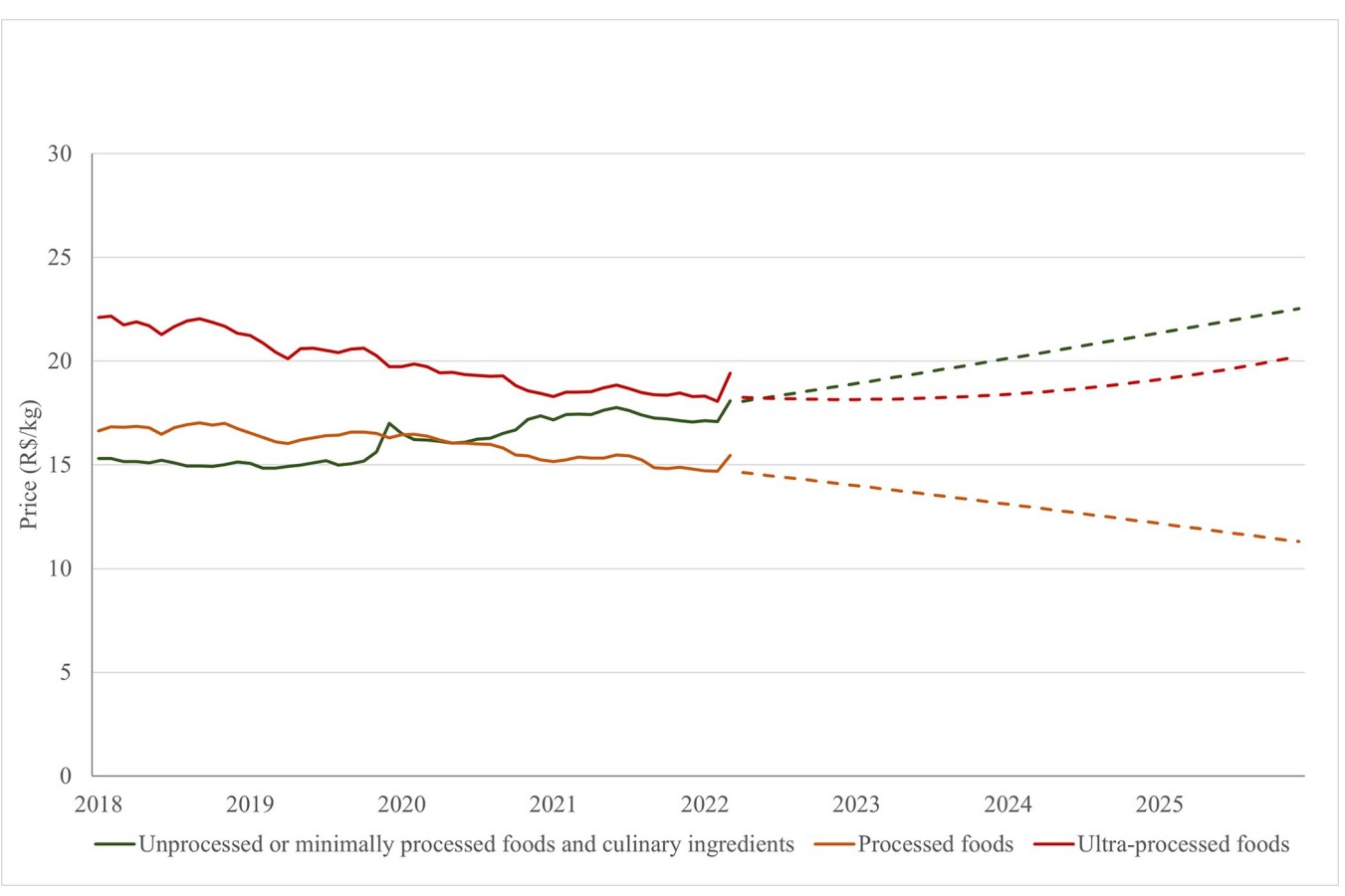

**Fig 1. Mean monthly price (R$/Kg) of unprocessed or minimally processed foods and processed culinary ingredients, processed food, and UPFP from January 2018 to March 2022 and forecast up to 2025.** Brazil, 2018–2025.

processed foods (from R$ 16.6/kg in 2018 to R$ 15.5/kg in March 2022) and UPFP (from R$ 22.1/kg in 2018 to R$ 19.4/kg in 2022), however, decreased in the same period.

Fig 1 shows the price of food groups from January 2018 to March 2022 and forecast up to 2025. Sustaining the trend observed in Table 1, the forecast for 2025 indicates a further increase in unprocessed or minimally processed foods and processed culinary ingredients prices (R$ 15.3/kg in 2018 to R$ 22.5/kg in 2025), a decrease in processed foods (R$ 16.6/kg in 2018 to R$ 11.33/kg in 2025) and UPFP prices (R$ 22.1/kg in 2018 to R$ 20.2/kg in 2025). There is also an intense increase in the price of fresh and minimally processed foods from 2020 onwards. According to these predictions, unprocessed or minimally processed foods will become more expensive them UPFP and processed food in Brazil from 2022.

Fig 2 shows the relative price of healthy foods (unprocessed or minimally processed foods and processed culinary ingredients) in relation to unhealthy foods (UPFP). Between January 2018 and March 2022, the relative price of healthy to unhealthy foods increased from 69.3% to 94.6%. In May 2022, this ratio exceeds 100.00%, reaching 130.8% in 2025.

Fig 3 shows the accumulated values of the IPC of food groups from January 2018 to March 2022. All food groups showed an increase in the IPC in this period, although this growth was less pronounced and more continuous between processed foods and UPFP. The index of unprocessed or minimally processed foods and processed culinary ingredients had higher variations over the period, with a peak in 2018 followed by a period of stability until 2019. In 2020,

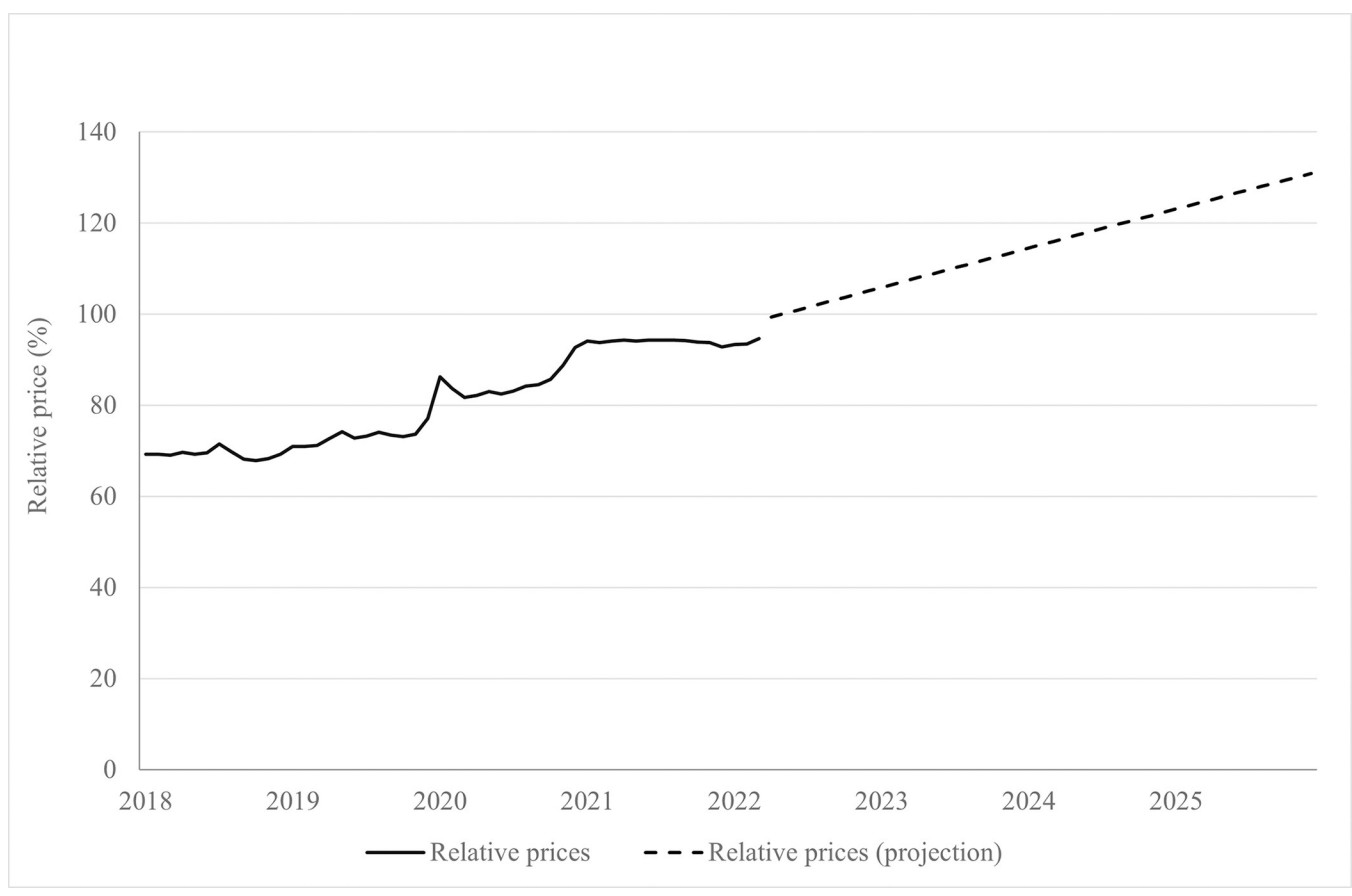

**Fig 2. Price of unprocessed or minimally processed foods and processed culinary ingredients relative to the price of UPFP (%) from January 2018 to March 2022 and forecast up to 2025.** Brazil, 2018–2025.

the IPC returned to growth. At the end of 2019, the IPC of healthy foods surpassed that IPC of unhealthy foods, reaching a 40-percentage points difference at the end of the studied period.

## Discussion

In Brazil, up to the moment of the conclusion of the present study (1st semester of 2022) healthy foods (unprocessed or minimally processed foods and processed culinary ingredients) were less expensive than unhealthy foods (processed foods and UPFP). However, the analyses from the present study, involving data from January 2018 to March 2022, suggest the reversal of this food price pattern. Projection analyzes made before the Covid-19 pandemic (involving data from January 1994 to December 2017) already predicted this inversion, but forecasted it to 2026 [12].

This change in the price trends initially predicted for Brazil seems to reflect the impact of the Covid-19 pandemic on the global economy. The disorganization of globalized production chains turned the pandemic an inflationary event worldwide [26]. Regarding specifically food products, export/import restrictions, along with restrictive measures (social distancing and isolation), interfered in food supply, affecting both producers and consumers and generating insufficient production and flow [15].

Further, while aggressive monetary policy in high-income countries increased the demand for certain foods (such as meat and other animal products), insufficient social security actions

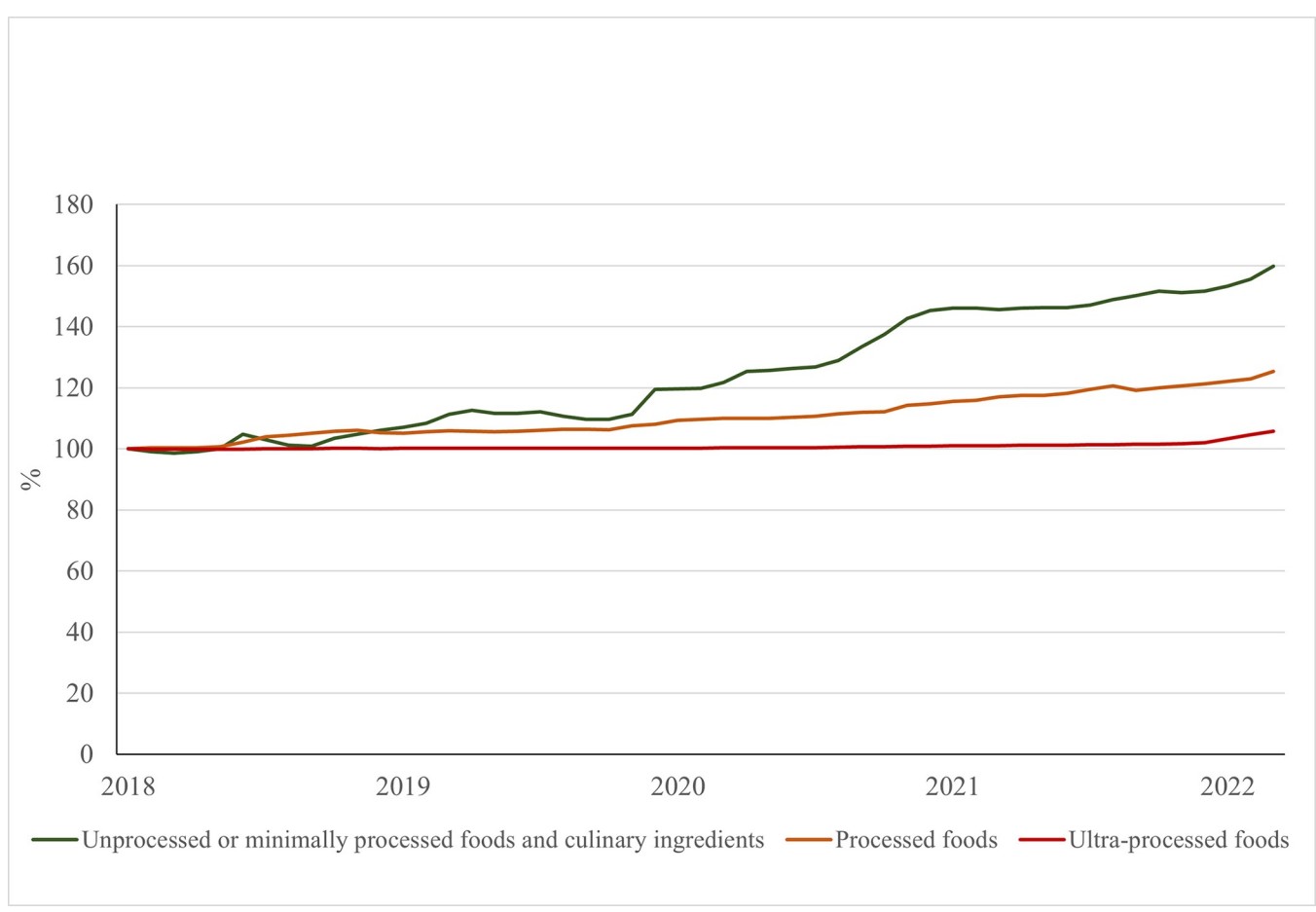

**Fig 3. Accumulated values of the IPC of unprocessed or minimally processed foods and processed culinary ingredients, processed foods, ultra-processed food, and food consumption at home from January 2018 to March 2022 and forecast up to 2025.** Brazil, 2018–2025.

in low and middle income impacted food consumption patterns [15,26]. This scenario with restricted offers and increased demand negatively reflects on food prices [14–16]. Restrictive measures have been reported to increase food prices and price inflation dispersion worldwide, including in Brazil [15–18,27].

The reduction in the UPFP price during 2018–2019 is coherent with prior estimations [12]. However, the continuity of this trend during 2020, as opposed to the increase in the price of unprocessed or minimally processed foods, is, for sure, surprising. Although our study does not provide the causes of the observed trends, some hypothesis might include the use of inventories and hedging contracts to purchase commodity ingredients. Both strategies are widely employed in food industry while mostly unavailable for small producers of unprocessed or minimally processed foods.

This trend further encourages replacing traditional meals with unhealthy foods, imposing a health risk to the population. As shown by the literature, the consumption of UPFP is associated with a decline in the diet's nutritional profile [28] and with health outcomes such as obesity, metabolic syndrome, diabetes, and different types of cancer [29–36].

The increases in the price of unprocessed or minimally processed foods and processed culinary ingredients, especially during the Covid-19 pandemic (from 2020 onwards), exacerbates the food insecurity in Brazil. Before the Cvoid-19 pandemic, the Brazilian population was

already in extreme vulnerability, with high unemployment rates and disruption of social policies [37]. However, this scenario was worsened during the Covid-19 pandemic due to work instability and decreased family income.

The economic crisis has affected access to food, increasing the risk of food insecurity [38–40]. Data from the "National survey on food insecurity in the context of the Covid-19 pandemic in Brazil" showed that the pandemic accelerated the increase in the prevalence of food insecurity in the country, which already had growth projections since 2013. Between 2018 and 2020, mild food insecurity increased from 20.7% to 34.7%, moderate food insecurity increased from 10.1% to 11.5%, and severe food insecurity increased from 5.8% to 9.0% [40]. Currently in Brazil, 125.2 million Brazilians are food insecure and more than 33 million are hungry [41].

The persistent increase in food inflation further compounds the issue of food insecurity in the nation. Fig 2 serves as a clear depiction of this scenario, showcasing two distinct peaks of inflation. The first peak, occurring at the end of 2019, primarily impacts the prices of minimally processed foods. This initial surge likely stems from various factors, including disruptions in supply chains and market uncertainties preceding the onset of the pandemic. The subsequent peak, observed in 2022, extends its impact across all food categories, exacerbating the challenges already posed by the pandemic.

Brazil faced a complex scenario, marked by an economic and health crisis. The pandemic highlighted social inequalities, and the crisis underscored flaws in the neoliberal model. The human right to healthy and adequate food has been neglected by the Federal Government even before the onset of the pandemic. Neoliberal policies have contributed to the diminishing role of the State in ensuring food security and nutrition, resulting in the weakening of distribution centers, public food stocks, the Food Acquisition Program (*Programa de Aquisição de Alimentos* (PAA)), and agrarian reform initiatives. This trend was further underscored by the abrupt abolition of the National Council for Food and Nutritional Security (*Conselho Nacional de Segurança Alimentar e Nutricional*—Consea) on the first day of Jair Bolsonaro's government [42].

Now, government interventions are even more necessary to ensure the population's food security and curb the advance in the prevalence of obesity and other non-communicable diseases (NCD) in the country. In this sense, conditional cash transfer programs (CCTP) seem to be an action capable to ensuring short-term results. Brazil already has a CCTP, named *Bolsa Família* program, since 2003 [43]. This program was recently renamed, *Auxílio Brasil*. During the Covid-19 pandemic (2021–2022), the program was reorganized [44]. Some conditionalities were removed while the maximum amount possible to be received by each beneficiary family increased after three years. A comprehensive study has been conducted to determine if the current format can in fact be effective.

Longer-term measures aiming to decrease the price of healthy foods and increase the price of the unhealthy ones are also necessary. Such measures could include better incentives for the production and commercialization chain of healthy foods, such as tax exemptions and financing with fees below market values [45,46]. Futhermore, taxation of unhealthy foods is also an important possibility [45]. For instance, the taxation of sugary drinks has been adopted in different countries and directly impacted the consumption of these products [7–9].

UPFP are associated with intensive agriculture/livestock, mainly through intensive monocultures, production of large amounts of inputs and loss of biodiversity. Combining large-scale use of low-cost ingredients and increased consumption worldwide affects all dimensions of the food system. Only reducing the intake of energy dense UPFP could aid a substantial reduction in greenhouse gas emissions (GGE). In other words, reducing the consumption of UPFP can enhance the food system's sustainability and contribute to better health conditions (reduction of NCD) [47].

The present study has some limitations. Various methods of measuring food prices can yield conflicting results regarding the affordability of healthy diets [48]. Prices per unit of energy are often significantly affected by the energy density of foods, leading to data that can be challenging to interpret, particularly for low-calorie foods and beverages. Therefore, to mitigate this bias and offer insights beyond the nutritional standpoint, we computed actual price series based on price per unit of weight (R$/kg). The use of price per kg presents stability in price fluctuations when we observe temporal trends [12].

While the price of certain minimally processed foods, like fruits and vegetables, tends to fluctuate more than culinary ingredients, such as oils and sugar, we chose to evaluate these food groups together. This decision was made because the consumption of these food groups is usually associated with and represents a dietary pattern.

The number of foods and beverages included in the study was small (n 95) and only reflects those in the IPCA list rather than a full range of items available on the Brazilian market. Nonetheless, the items included in the analysis reflect the most purchased by the Brazilian population (approximately two-thirds of the total energy acquired), considering the role of the IPCA as a measure of consumer inflation.

Therefore, the IPC structure used to select most purchased prices and to define weighs for product aggregation was stablished in the most recent POF (2017/18), before the Covid-19 pandemic, and may be vulnerable to subtle changes in consumer behavior. A final limitation concerns product aggregation in the IPCA database, which made it impossible to produce individualized estimates for each of the products. IBGE states that only products with similar inflation behavior are aggregated, although this is not verifiable thorough the data available.

## Author Contributions

**Conceptualization:** Giovanna Calixto Andrade, Thaís Cristina Marquezine Caldeira, Laís Amaral Mais, Ana Paula Bortoletto Martins, Rafael Moreira Claro.

**Formal analysis:** Thaís Cristina Marquezine Caldeira.

**Investigation:** Giovanna Calixto Andrade.

**Supervision:** Rafael Moreira Claro.

**Writing – original draft:** Giovanna Calixto Andrade, Thaís Cristina Marquezine Caldeira.

**Writing – review & editing:** Giovanna Calixto Andrade, Thaís Cristina Marquezine Caldeira, Laís Amaral Mais, Ana Paula Bortoletto Martins, Rafael Moreira Claro.

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
