## [Decision Letter · Decision Letter 0]

8 Feb 2024

PONE-D-23-30675Food price trends during COVID-19 pandemic in BrazilPLOS ONE

Dear Dr. Andrade,

Thank you for submitting your manuscript to PLOS ONE. After careful consideration, we feel that it has merit but does not fully meet PLOS ONE’s publication criteria as it currently stands. Therefore, we invite you to submit a revised version of the manuscript that addresses the points raised during the review process.

We look forward to receiving your revised manuscript.

Kind regards,

Mathias Roberto Loch, Ph.D

Academic Editor

PLOS ONE

Journal Requirements:

4. Thank you for stating the following in the Acknowledgments Section of your manuscript: "We thank the Coordenação de Aperfeiçoamento de Pessoal de Nível Superior – Brasil(CAPES), the Conselho Nacional de Desenvolvimento Científico e Tecnológico (CNPq))and the International Development Research Centre (IDRC) for financial support."

Please remove any funding-related text from the manuscript and let us know how you would like to update your Funding Statement. Currently, your Funding Statement reads as follows: "Coordenação de Aperfeiçoamento de Pessoal de Nível Superior – Brasil (CAPES)

Conselho Nacional de Desenvolvimento Científico e Tecnológico (CNPq)

International Development Research Centre (IDRC)"

5. Thank you for stating the following in your Competing Interests section: "none"

Additional Editor Comments:

Dear authors,

Two revisions to the manuscript follow.

In a complementary way, I suggest that the authors minimally address the way in which the Brazilian government acted during the pandemic, as I believe that food prices also partly reflect this issue, including the neoliberal approach of the Bolsonaro government.

Reviewers' comments:

Reviewer's Responses to Questions

**Comments to the Author**

1. Is the manuscript technically sound, and do the data support the conclusions?

Reviewer #1: Yes

Reviewer #2: Yes

2. Has the statistical analysis been performed appropriately and rigorously? 

Reviewer #1: I Don't Know

Reviewer #2: Yes

3. Have the authors made all data underlying the findings in their manuscript fully available?

Reviewer #1: Yes

Reviewer #2: Yes

4. Is the manuscript presented in an intelligible fashion and written in standard English?

Reviewer #1: Yes

Reviewer #2: No

5. Review Comments to the Author

Reviewer #1: Thank you for the study of this very important topic. It is appreciated that this is a challenging area to study, often limited by the data available. In general, the study requires some editing of English language and expression, in order to better convey the authors’ findings. Some specific instances have been identified below, together with other comments.

Abstract

The meaning of the phrase “However, the analyses suggested the reversal of this price pattern” is unclear given the previous sentence is stating results of the current study. Please re-phrase.

Methods

Initially, the set of 95 food and beverage items are stated as being the most consumed in the country, but later it is stated that these items are those in the IPCA which have sufficient data and are not infusions or alcohol. It is not clear whether the IPCA items with sufficient data are indeed the most consumed.

The phrase “ranging from 1 to 40 minimum wages” is not clear. Does it mean 1 to 40 times the minimum wage?

Data organization

The second sentence states “to calculate prices from 2018 to 2021” yet elsewhere it is stated prices were calculated to 2022.

The price calculation is very confusing. Is A in formula 1 the same as G in formula 2? Similarly, are B & C the same as E & F? Are these two formulae combined to give the prices of Table 1 and the Figures?

Results

Why does Figure 1 show a sudden uptick in price/kg in 2022? Can the sudden rise in price of unprocessed foods etc in 2020 be explained?

I think there should be some discussion of the use of price/kg as the metric. Processed culinary ingredients include foods that have a high price/kg e.g. oils, but which are used in small amounts in the diet. Thus combining these prices with unprocessed foods such as vegetables that have a much lower price/kg but have much fewer calories, and are used in larger amounts, seems like it will bias the results. Consider the work of Carlson et al (Carlson A, Frazao E. Are Healthy Foods Really More Expensive? It Depends on How You Measure the Price. United States Department of Agriculture Economic Research Service; 2012.)

Simplification of the analysis to just three groups of foods does lose a lot of nuance of the differing impacts on the price of different types of food. For example, local weather events can influence fruit and vegetable prices, but larger world events can influence price of foods such as oils (price of canola oil has increase due to the war in Ukraine, for example). Some acknowledgement of this should be included.

Reviewer #2: Evaluation of paper: Food price trends during COVID-19 pandemic in Brazil

Manuscript Number: PONE-D-23-30675

The article refers to the evolution of food prices in Brazil from 2020 to 2022 (Covid-19 pandemic) and the price scenario projects until 2025, considering the type of food processing.

The text is well written and has scientific and political relevance in Brazil, because the analyzes and reflections presented. Bibliographic references are updated because that 31/46 refer to the last five years.

However, I highlight some aspects that need improvement:

Results Section:

Table 1:

- Add location and analyzed period;

- Include the unit of measurement R$/kg in the column headers;

- I suggest highlighting in bold the values referring to the main groups by type of processing;

Figure 1:

- It does not have good resolution and sharpness.

Figure 2

- It is not possible to visualize in the graph presented what is described in the title and text;

- The title of figure 2 is incomplete in the presentation of the graph;

- The image does not have good resolution and sharpness;

Figure 3

- The image does not have good resolution and sharpness;

- A new food group not highlighted in any other table or illustration appears, just as it was not described in the methodology: “Food Consumption at Home”. I suggest removing this group.

6. PLOS authors have the option to publish the peer review history of their article (what does this mean?). If published, this will include your full peer review and any attached files.

Reviewer #1: No

Reviewer #2: **Yes: **Clelia de Oliveira Lyra, Associate Professor, Department of Nutrition, Federal University of Rio Grande do Norte

---

## [Author Response · Author response to Decision Letter 0]

12 Apr 2024

Dear Editor,

We appreciate the review carried out and the contribution to improving the manuscript (PONE-D-23-30675). We are available to provide any additional information if necessary. We hope to have responded to all the reviewers' suggestions and comments, and we await your decision.

To enhance the article's quality, the entire text has been reviewed by a company specializing in English translation and editing.

Company Information:

 Name: Lira Traduções e Revisões

 Phone: +55 (16) 9 9352 8737

 E-mail: liratraducoes@gmail.com

Website: https://liratraducoes.com/

Best regards,

Authors

 

Comments to the Author:

Reviewer #1: Thank you for the study of this very important topic. It is appreciated that this is a challenging area to study, often limited by the data available. In general, the study requires some editing of English language and expression, in order to better convey the authors’ findings. Some specific instances have been identified below, together with other comments.

We appreciate the points raised, we have revised the text according to the suggestions. The text has been reviewed by experts to ensure proper English language.

Abstract

The meaning of the phrase “However, the analyses suggested the reversal of this price pattern” is unclear given the previous sentence is stating results of the current study. Please re-phrase.

We have improved the abstract in line with the suggestions.

“The present study aims to analyze the trends in food price in Brazil with emphasis on the period of the COVID-19 pandemic (from March 2020 to March 2022). Data from the Brazilian Household Budget Survey and the National System of Consumer Price Indexes were used as input to create a novel data set containing monthly prices (R$/Kg) for the foods and beverages most consumed in the country between January 2018 and March 2022. All food items were divided according to the NOVA food classification system. We estimated the mean price of each food group for each year of study and the entire period. The monthly price of each group was plotted to analyze changes from January 2018 to March 2022. Fractional polynomial models were used to synthesize price changes up to 2025. Results of the present study showed that in Brazil unprocessed or minimally processed foods and processed culinary ingredients were more affordable than processed and ultra-processed foods. However, trend analyses suggested the reversal of the pricing pattern. The anticipated changes in the prices of minimally processed food relative to ultra-processed food, initially forecasted for Brazil, seem to reflect the impact of the COVID-19 pandemic on the global economy. These results are concerning as the increase in the price of healthy foods aggravates food and nutrition insecurity in Brazil. Additionally, this trend encourages the replacement of traditional meals for the consumption of unhealthy foods, increasing a health risk to the population.”

Methods

Initially, the set of 95 food and beverage items are stated as being the most consumed in the country, but later it is stated that these items are those in the IPCA which have sufficient data and are not infusions or alcohol. It is not clear whether the IPCA items with sufficient data are indeed the most consumed.

We apologize for the lack of clarity in the text. The excerpts were modified to reinforce the explanation that the data collected in the SNIPC are derived from the POF, therefore, the items are similar to the most consumed food data collected. As for infusions and alcoholic drinks items, a more in-depth explanation about their removal has been added.

“The SNIPC, implemented and managed by IBGE, continuously and systematically calculates the Consumer Price Index (Índice de Preços ao Consumidor - IPC). This index aims to identify the oscillation in the prices of goods and services related to the basket of goods in the Brazilian population. We defined the consumption baskets, with items consumed in the country, and the update of the IPC/SNIPC weighting structures through information from POF, which is carried out in the country, portraying the diversity of consumption habits observed throughout the Brazilian territory [19]. We used the Extended Consumer Price Index (Índice Nacional de Preços ao Consumidor Amplo - IPCA). The IPCA aims to measure the inflation of retail products and services related to the personal consumption of Brazilian with monthly incomes ranging from 1 to 40 minimum wages (an income range that guarantees the coverage of 90% of families belonging to urban areas covered by SNIPC, regardless of the source of income) [21].”

The phrase “ranging from 1 to 40 minimum wages” is not clear. Does it mean 1 to 40 times the minimum wage?

We have included in the text an explanation on the use of the term “variation from 1 to 40 minimum wages”. This income range is used with the aim of guaranteeing coverage of 90% of families belonging to urban areas covered by the SNIPC.

“The IPCA aims to measure the inflation of retail products and services related to the personal consumption of Brazilian with monthly incomes ranging from 1 to 40 minimum wages (an income range that guarantees the coverage of 90% of families belonging to urban areas covered by SNIPC, regardless of the source of income) [21].”

Data organization:The second sentence states “to calculate prices from 2018 to 2021” yet elsewhere it is stated prices were calculated to 2022.

We apologize for the mistake. The text has been corrected for March 2022.

“The SNIPC does not provide proper price data, only monthly price changes (by IPCA). Thus, we selected unit prices from POF 2017/2018 to calculate prices from 2018 to March 2022, using the monthly variation of the IPCA.”

The price calculation is very confusing. Is A in formula 1 the same as G in formula 2? Similarly, are B & C the same as E & F? Are these two formulae combined to give the prices of Table 1 and the Figures?

The formulas are complementary, formula 1 shows how we calculated the nominal price values and formula 2, how the real prices were calculated based on formula 1. We reinforce the items referred to in each formula in the text. A and G are similar, we changed the text to represent the same item as formula 1. B and C differ from E and F. We tried to make this clearer in the classification of each item. The two formulas lead to the values identified for each month/year presented in the tables and figures. 

Results

Why does Figure 1 show a sudden uptick in price/kg in 2022? Can the sudden rise in price of unprocessed foods etc in 2020 be explained?

The peak in 2020 was due to a large increase in the price of fresh and minimally processed foods during the Covid 19 pandemic, these prices continued to increase due to inflationary factors in the country, including at the beginning of 2022. We have included this caveat in the text about price increases during the period.

“The persistent increase in food inflation further compounds the issue of food insecurity in the nation. Fig 2 serves as a clear depiction of this scenario, showcasing two distinct peaks of inflation. The first peak, occurring at the end of 2019, primarily impacts the prices of minimally processed foods. This initial surge likely stems from various factors, including disruptions in supply chains and market uncertainties preceding the onset of the pandemic. The subsequent peak, observed in 2022, extends its impact across all food categories, exacerbating the challenges already posed by the pandemic.”

I think there should be some discussion of the use of price/kg as the metric. Processed culinary ingredients include foods that have a high price/kg e.g. oils, but which are used in small amounts in the diet. Thus combining these prices with unprocessed foods such as vegetables that have a much lower price/kg but have much fewer calories, and are used in larger amounts, seems like it will bias the results. Consider the work of Carlson et al (Carlson A, Frazao E. Are Healthy Foods Really More Expensive? It Depends on How You Measure the Price. United States Department of Agriculture Economic Research Service; 2012.)

We agree with the points raised. Indeed, the chosen metric for assessing food prices directly influences the results and their interpretation. However, we opted for the use of price/kg instead of price/calories to prevent the energy density of foods (especially items with no calories, such as diet sodas) from influencing the findings. Furthermore, this metric allows for less variability in data when we analyze temporal trends. We have included this limitation of the study in the discussion.

“The present study has some limitations. Various methods of measuring food prices can yield conflicting results regarding the affordability of healthy diets [47]. Prices per unit of energy are often significantly affected by the energy density of foods, leading to data that can be challenging to interpret, particularly for low-calorie foods and beverages. Therefore, to mitigate this bias and offer insights beyond the nutritional standpoint, we computed actual price series based on price per unit of weight (R$/kg). The use of price per kg presents stability in price fluctuations when we observe temporal trends [12].”

Simplification of the analysis to just three groups of foods does lose a lot of nuance of the differing impacts on the price of different types of food. For example, local weather events can influence fruit and vegetable prices, but larger world events can influence price of foods such as oils (price of canola oil has increase due to the war in Ukraine, for example). Some acknowledgement of this should be included.

The decision to evaluate minimally processed foods with culinary ingredients aims to understand the price of diets based on real dietary patterns, as the consumption of culinary ingredients is almost always associated with the consumption of minimally processed foods. We have included this limitation and information in the manuscript.

“While the price of certain minimally processed foods, like fruits and vegetables, tends to fluctuate more than culinary ingredients, such as oils and sugar, we chose to evaluate these food groups together. This decision was made because the consumption of these food groups is usually associated with and represents a dietary pattern.”

 

Reviewer #2: 

The article refers to the evolution of food prices in Brazil from 2020 to 2022 (Covid-19 pandemic) and the price scenario projects until 2025, considering the type of food processing.

The text is well written and has scientific and political relevance in Brazil, because the analyzes and reflections presented. Bibliographic references are updated because that 31/46 refer to the last five years.

However, I highlight some aspects that need improvement:

Results Section:

Table 1:

- Add location and analyzed period;

- Include the unit of measurement R$/kg in the column headers;

- I suggest highlighting in bold the values referring to the main groups by type of processing;

Figure 1:

- It does not have good resolution and sharpness.

Figure 2

- It is not possible to visualize in the graph presented what is described in the title and text;

- The title of figure 2 is incomplete in the presentation of the graph;

- The image does not have good resolution and sharpness;

Figure 3

- The image does not have good resolution and sharpness;

- A new food group not highlighted in any other table or illustration appears, just as it was not described in the methodology: “Food Consumption at Home”. I suggest removing this group.

We appreciate the feedback. We have made changes to Figures 2 and 3 to ensure that all elements appear clearly in the images. Regarding the resolution of the images, we believe that the PDF generated during submission may degrade the image quality, but when the figures are downloaded from the document itself, they appear with good resolution.

---

## [Editor Report · Decision Letter 1]

1 May 2024

Food price trends during the COVID-19 pandemic in Brazil

PONE-D-23-30675R1

Dear Dra Andrade,

We’re pleased to inform you that your manuscript has been judged scientifically suitable for publication and will be formally accepted for publication once it meets all outstanding technical requirements.

Kind regards,

Mathias Roberto Loch, Ph.D

Academic Editor

PLOS ONE
---

## [Editor Report · Acceptance letter]

14 May 2024

PONE-D-23-30675R1 

PLOS ONE

Dear Dr. Andrade, 

I'm pleased to inform you that your manuscript has been deemed suitable for publication in PLOS ONE. Congratulations! Your manuscript is now being handed over to our production team.

Kind regards, 

on behalf of

Dr. Mathias Roberto Loch 

Academic Editor

PLOS ONE